# Assessment of Humoral and Long-Term Cell-Mediated Immune Responses to Recombinant Canarypox-Vectored Equine Influenza Virus Vaccination in Horses Using Conventional and Accelerated Regimens Respectively

**DOI:** 10.3390/vaccines10060855

**Published:** 2022-05-26

**Authors:** Charles El-Hage, Carol Hartley, Catherine Savage, James Watson, James Gilkerson, Romain Paillot

**Affiliations:** 1Centre for Equine Infectious Diseases, Faculty of Veterinary and Agricultural Sciences, The University of Melbourne, Parkville, VIC 3010, Australia; carolah@unimelb.edu.au (C.H.); katesav@me.com (C.S.); jrgilk@unimelb.edu.au (J.G.); 2Australian Centre for Disease Preparedness, CSIRO, Geelong, VIC 3216, Australia; james.watson@csiro.au; 3School of Equine and Veterinary Physiotherapy, Writtle University College, Lordship Road, Writtle, Chelmsford CM1 3RR, UK; romain.paillot@writtle.ac.uk

**Keywords:** equine influenza, humoral, cell-mediated, interferon-gamma, antibody, cross protection, H3N8, rCP-EIV, ProteqFlu™/ProteqFlu^©^-TE

## Abstract

During Australia’s first and only outbreak of equine influenza (EI), which was restricted to two northeastern states, horses were strategically vaccinated with a recombinant canarypox-vectored vaccine (rCP-EIV; ProteqFlu™, Merial P/L). The vaccine encoded for haemagglutinin (HA) belonging to two equine influenza viruses (EIVs), including an American and Eurasian lineage subtype that predated the EIV responsible for the outbreak (A/equine/Sydney/07). Racehorses in Victoria (a southern state that remained free of EI) were vaccinated prophylactically. Although the vaccine encoded for (HA) belonged to two EIVs of distinct strains of the field virus, clinical protection was reported in vaccinated horses. Our aim is to assess the extent of humoral immunity in one group of vaccinated horses and interferon-gamma ((EIV)-IFN-γ)) production in the peripheral blood mononuclear cells (PBMCs) of a second population of vaccinated horses. Twelve racehorses at work were monitored for haemagglutination inhibition antibodies to three antigenically distinct equine influenza viruses (EIVs) The EIV antigens included two H3N8 subtypes: A/equine/Sydney/07) A/equine/Newmarket/95 (a European lineage strain) and an H7N7 subtype (A/equine/Prague1956). Cell-mediated immune responses of: seven racehorses following an accelerated vaccination schedule, two horses vaccinated using a conventional regimen, and six unvaccinated horses were evaluated by determining (EIV)-IFN-γ levels. Antibody responses following vaccination with ProteqFlu™ were cross-reactive in nature, with responses to both H3N8 EIV strains. Although (EIV)IFN-γ was clearly detected following the in vitro re-stimulation of PBMC, there was no significant difference between the different groups of horses. Results of this study support reports of clinical protection of Australian horses following vaccination with Proteq-Flu™ with objective evidence of humoral cross-reactivity to the outbreak viral strain A/equine/Sydney/07.

## 1. Introduction

Equine influenza (EI) is a highly contagious respiratory disease of horses, considered to be the most common and important viral equine respiratory pathogen, causing serious and widespread epizootics worldwide [1,2]. Naïve horses are highly susceptible to infection, with clinical signs that typically include pyrexia, coughing, and nasal discharge [3,4,5]. The 2007 Australian EI outbreak resulted in over 70,000 horses infected on over 10,000 properties in two states [6,7]. It was estimated to have cost over AUD 1 billion for the measures associated with control and eradication. Severe disruption to equine pursuits resulted for several months, and although most horses recovered after mild to moderate illness, some deaths were recorded [7,8,9,10]. Vaccination provides a vital method of clinical protection for horses against EI [11,12]. An established correlation between vaccine-induced antibodies and clinical protection against homologous EIVs has been well recognised [13,14,15]. Despite this, clinical protection from experimental EIV challenge has been recorded in ponies in the absence of circulating antibodies [16]. It is presumed that cell-mediated immune responses, in addition to local mucosal factors, are responsible for clinical protection from EIV in such conditions [5,17,18]. Cell-mediated immunity following influenza infection invokes a range of responses that reduce the severity of infection and facilitate a more rapid recovery in several mammalian species [5,16,17,19]. 

The basic tenet of vaccination relies upon safe exposure of a host’s immune system to an antigen, triggering a range of immunological responses, including stimulating the production of memory cells [20,21,22]. Memory B- and T-lymphocytes mount an anamnestic immune response following re-exposure to a pathogen. Subsequent protection from disease thus results from rapid antigenic re-stimulation via clonal selection and expansion of memory T- and B-cell populations [20,21]. 

Cell-mediated immune (CMI) responses are considered important not only in limiting morbidity but also in enabling a degree of cross-protection between strains [5,23,24]. Whilst few studies have been performed on horses, cell-mediated responses in humans have been demonstrated to be directed to antigens conserved between strains, hence being cross-protective [5,16,17,25]. 

Induction of heterologous immune responses in horses to EIVs has practical significance as the constant antigenic drift of influenza virus strains results in outbreak strains that are potentially divergent from vaccine strains, reducing the efficacy of the existing immune response [4,26,27]. Assessment of CMI responses in horses is complex and not routinely conducted [24]. Production of IFN-γ, however, is considered a reliable and practical marker of a Type 1 cellular immune response [24,28]. In humans, cellular immune mechanisms are considered to play an important role in the clearance of influenza virus from the respiratory tract [25,29]. 

Evaluation of CMI responses in vaccinated horses has not been widely reported until the current century. The development of more sophisticated and novel vaccines, such as the recombinant canarypox-vectored, specifically adjuvanted and cold-adapted modified live EIV vaccines, has prompted the development of assays to investigate a wider range of responses, including mucosal IgA responses and CMI [23,24,28,30,31,32,33,34].

There are few reports of long-term CMI responses to vaccination or infection in horses [17,35]. One report of eight horses vaccinated between 4 to 11 years previously with Clostridium tetani toxoid (tetanus) detected enhanced responses in all horses following re-vaccination [36]. Although the nature of the immune response was not determined, this was likely to be a Type II (predominantly humoral) response [22], and antigenic stimulation through field exposure could not be ruled out. A recent study monitored humoral responses to an EI vaccine schedule 599 days after V1; however, a further three booster vaccinations had been given, the last being 14 days prior to sampling [37], although the authors of this study detected significantly increased IFN-γ production in ponies one year after vaccination (Dr R. Paillot, unpublished manuscript). Anecdotal reports, in addition to several reports from the Australian EI outbreak, indicated that clinical signs of disease in the few horses that had been previously vaccinated several years prior to the outbreak were comparably milder than in unvaccinated horses [9,38].

An accelerated EI vaccination programme has been suggested and utilised as part of a post-outbreak vaccination strategy [10,38,39,40]. There have been few studies, however, to investigate the immune response to this regimen, particularly in terms of CMI [39,41]. Previous work by this group has demonstrated similar SRH antibody responses in horses vaccinated with accelerated regimens to those published for conventional regimens [30,39,41,42].

The aim of this study is to examine immune responses in horses following vaccination with ProteqFlu™/ProteqFlu^©^-TE. The first part monitored humoral responses in racehorses, and the second part identified IFN-γ production, indicative of an anamnestic Type 1 immune response, in seven horses following an accelerated EIV vaccination regimen approximately 6 years prior. Since this work was conducted in a state that never recorded a case of EI [7], immune responses to EIV were exclusively due to vaccination, without possible contamination or exposure to circulating field strains.

## 2. Materials and Methods

### 2.1. Study Populations

The study populations were comprised of two groups of mature racehorses following vaccination with ProteqFlu™/ProteqFlu™-Te. In Group 1, the antibody responses were examined, and in Group 2, there was an evaluation of long-term CMI.

Previous vaccination history for each horse was not recorded; however, EIV vaccination in Australia is only permitted for export purposes. 

### 2.2. Animal Ethics

Many of the samples collected for this study were collected as part of the EI surveillance and monitoring program of the Chief Veterinary Officer’s department, Victoria. In addition, animal ethics approval for follow-up blood sampling was obtained under the Faculty of Veterinary Science Animal Ethics Committee (Ethics I.D. 1212481).

### 2.3. Statistical Analysis

The Mann–Whitney U-test for significance was applied to determine differences between haemagglutinin inhibition (HI) antibody responses; *p*-values less than 0.05 were considered significant. A mixed-effects regression model was utilised to determine differences between the production of equine influenza virus-specific IFN-γ levels following the re-stimulation of PBMC between groups of horses. 

### 2.4. Vaccination

The vaccine authorised for use during Australia’s EI outbreak was ALVAC^©^-EIV, a recombinant canarypox-vectored vaccine expressing the HA genes of two influenza H3N8 virus strains, A/equine/Kentucky/94 and A/equine/Newmarket/2/93, representing American and Eurasian lineages respectively. The vaccine was commercially available in Europe as ProteqFlu™/ProteqFlu™–Te (Merial P/L). The vaccines were administered in the neck by deep intramuscular injection. A one mL vial contained greater than 105.2 50% fluorescence assay infectious doses for each virus strain and was adjuvanted with carbomer 974P. ProteqFlu™–Te was most commonly used during the study; this EI vaccine included tetanus toxoid in the diluent.

### 2.5. Vaccination Schedules

Two vaccination schedules were used in this study for the administration of the EI vaccines. An accelerated vaccination schedule was used in the CMI study, with the first (V1) and second (V2) vaccination intervals (reduced from 21 to 4) of 2 to 14 days, with the third vaccination (V3) administered 3 months later. The conventional vaccination schedule used for the humoral cross-reactivity study followed those recommended by the manufacturer, where V1–V2 were separated by 28 days. No booster (V3) was administered as EIV was no longer circulating and the outbreak had been controlled by that time.

### 2.6. Equine Influenza Viruses

Seven EIVs (H3N8 and H7N7 subtypes) were used in this study (Table 1), including the Australian outbreak strain A/equine/Sydney/07 (H3N8, Fc1).

### 2.7. Blood Collection

Whole-blood samples were collected from each horse by jugular venepuncture into 10 mL plain glass vacuum tubes (Becton Dickinson, Franklin Lakes, NJ, USA). When samples were taken on the day of vaccination (days 0, 14 and 105), blood was collected prior to injection. Following clotting, serum was removed and stored at –70 °C until analysis.

### 2.8. Peripheral Blood Mononuclear Cell Separation

Plasma was aspirated from heparinised blood (100 mL per horse) following centrifugation (1000× *g*, 7 min, 4 °C) prior to transport on ice. Purification of PBMC was performed by centrifugation on Ficoll-Paque™ PLUS (GE Healthcare Biosciences, Uppsala, Sweden) according to the manufacturer’s instructions. Cells were counted using a haemocytometer and trypan blue exclusion.

### 2.9. Haemagglutination Inhibition Antibody Assay

The haemagglutination inhibition (HI) test was performed in microtitre plates according to standard procedures (OIE, 1996). Assays were conducted utilising three EI viruses, as described in Table 1. Titres are expressed as the reciprocal of the minimum dilution required to completely inhibit haemagglutination. The initial dilution for all HI assays was 1 in 8.

### 2.10. Competitive Enzyme-Linked Immunosorbent Assay

Competitive enzyme-linked immunosorbent assays (c-ELISA), targeting antibodies to influenza virus A nucleoprotein, were performed on all samples at the Australian Centre for Disease Preparedness, CSIRO, Geelong, Victoria. Test samples that do not contain nucleoprotein-specific antibodies do not inhibit the signal in this assay. A level of inhibition of less than 40% relative to that of the positive control was considered negative [43]. Since vaccination with ProteqFlu™–Te only elicits antibodies to the HA component of EIVs, sera from horses that tested positive were considered likely to have been infected [43].

### 2.11. Measurement of Interferon-Gamma Production

Given the expected low frequency of EIV-specific T-lymphocytes in the blood of vaccinated and sampled horses, techniques have been described to facilitate the clonal expansion of antigen-specific memory cells [24]. Details of techniques are based upon those previously described (by author R.P.) [30,44]. 

## 3. Results

### 3.1. Study Population 1

Mature racehorses of both sexes (including geldings) comprised the study population. Horses were vaccinated according to the manufacturer’s recommendations by racetrack practitioners. Blood samples were collected by jugular venepuncture into ten mL plain glass vacuum tubes (Vacutainer^®^, Becton Dickinson, Franklin Lakes, NJ, USA) on or about days 0, 14, 28, 56 and 84 (three additional samples were taken on day 112)in the weeks following the vaccinations. If vaccination occurred on the same day as venepuncture, the blood samples were collected first. Of the racehorses initially enrolled in the study and tested at day zero, sera from twelve horses were selected for inclusion in this study, having been sampled on at least four occasions. These twelve horses were from four different stables serviced by two multi-person veterinary racetrack practices (Table 2).

### 3.2. Serologic Testing

#### 3.2.1. Competitive ELISA Assay

All samples collected before and after vaccination were negative for antibodies to influenza A virus nucleoprotein (NP) using the c-ELISA assay (data not shown). The lack of an antibody response to this NP provided evidence of a lack of exposure to field EIVs. This was important in terms of ensuring that all immune responses relating to EIVs were the sole result of vaccination. 

#### 3.2.2. HI Antibody Response to Vaccination

Haemagglutinin inhibition antibody responses to the two H3NA and the H7N7 subtype EIVs are indicated below. There was no significant difference between mean HI antibody levels to the H3N8 EIVs, A/equine/Sydney/07 and A/equine/Newmarket/95, representing the Eurasian and American lineages, respectively. Using the Mann–Whitney U-test for significance, the respective *p*-values were > 0.99, *p* = 0.44, *p* = 0.54 and *p* = 0.29 at 2, 4, 6 and 8 weeks, respectively. One horse (F20) failed to mount a detectable response to either antigen.

#### 3.2.3. A/Equine/Sydney/07 (H3N8)

Initial HI antibody responses appeared to be slow, with six horses responding minimally at day 14 and five horses recording no detectable HI antibody response. However, by day 42 (V2 plus 14 days), mean and individual peak antibody titres were highest at this time point (8 to 128). Results are summarised in Table 3 and Figure 1. 

#### 3.2.4. A/Equine/Newmarket/95(H3N8)

Haemagglutination inhibition antibody responses were similarly slow to rise against A/equine/Newmarket/95 (H3N8, Eurasian lineage). Responses remained fairly modest until day 42 (Table 3 and Figure 1). Maximal mean and individual highest HI titres were elicited at 56 days (V2 plus 28 days (range <8 to 128)). 

#### 3.2.5. A/Equine/Prague/1956(H7N7)

Sera from horses at all time points failed to inhibit haemagglutination to A/equine/Prague/1956 (H7N7), with no titres greater than 8 detected from any sample.

### 3.3. Group 2; Study Population

Fifteen thoroughbreds were employed in this study; nine had been vaccinated six years previously with ProteqFlu™-Te. Seven horses were part of an accelerated primary course EIV vaccination regimen, and two were vaccinated utilising a conventional primary course of 4 weeks. Six horses were included as unvaccinated breed-matched controls. All bar one conventionally vaccinated horse (Horse 16) were from Victoria, a state that has never recorded a case of EIVs. 

For the purposes of test positive controls, archived PBMCs were accessed from three ponies two weeks after infection with A/equine/Richmond/1/07 (H3N8, Fc 2 lineage) and stored in LN^2^ (Table 4). 

### 3.4. Interferon-Gamma Production 

Equine influenza virus-specific IFN-γ levels produced by PBMCs are illustrated in Figure 2. Although the median of vaccinates was numerically higher compared to control horses (0.07 and 0.03, respectively), the difference was not significant (*p* = 0.7; mixed-effects regression model). A statistically significant (*p* < 0.001) increase in IFN-γ production by PBMCs from experimentally infected (positive control) ponies was observed compared to control horses. 

## 4. Discussion

The majority of horses tested in this study population produced HI antibodies to two H3N8 virus strains following vaccination with ProteqFlu™-Te. Antibodies to A/equine/Sydney/07(H3N8), an American lineage Fc-1 virus, indicated cross-reactivity between vaccine and field strains and, hence, were suggestive of some protection against clinical disease. This EIV was responsible for the 2007 EI outbreak in Australia and was not included in commercial EI vaccines at the time, nor were any strains belonging to the Florida sublineage [7,8]. The American lineage strain encoded in ProteqFlu-Te was A/equine/Kentucky/94, which is phylogenetically and antigenically distant from the Florida sublineage viruses [45,46], and ProteqFlu currently encodes for the HA of Clades 1 and 2 viruses of the Florida sublineage (Merial datasheet).

Peak mean HI antibody titres to both H3N8 viruses were recorded 2–4 weeks following V2, consistent with the expected onset of protective/peak immunity [47,48]. The vaccine encoded A/equine/Kentucky/94 HA, an American lineage EIV that predates the Florida sublineage. Hence, the American lineage viruses were less closely related than the vaccine versus assay European lineage strains [11,45,49].

There are approximately 17 amino acid differences between the HA1 molecules of A/equine/Kentucky/94 and A/equine/Sydney/07, with 8 located in the antigenic sites identified in H3 influenza viruses [4,45,49]. As few as four amino acid changes, located in two separate antigenic sites, represent a significant antigenic drift for the human influenza virus in terms of immunological clinical protection [50]. The antigenic and genetic differences between A/equine/Sydney/07 and A/equine/Kentucky/94 are sufficiently diverse to compromise cross-protection and indeed fulfil the criteria required to update vaccine strains [51,52].

Since the HA of the European lineage vaccine virus (A/equine/Newmarket/2/93) is more closely related to the assay’s A/equine/Newmarket/95, a stronger antibody response may have been expected to the more divergent American sublineage viruses [11,53]. Despite these differences in HA within lineages, there was no statistical difference between mean HI antibody titres at any time point between the two lineages. 

Although the mean amplitude was considered low in this study, evidence of a heterologous antibody response was obvious. This finding was supported by reports of a reduction in the severity of clinical signs of disease in horses vaccinated during the 2007 Australian EI outbreak with ProteqFlu™/ProteqFlu™-Te. A report of amelioration in clinical signs of disease in horses that had been vaccinated recently prior to infection compared to unvaccinated horses at a nearby racetrack provided support for this view [9,38]. 

Antibodies to A/equine/Prague/1956 (H7N7) were not detected in any horses; this subtype has not been isolated since 1979 [11,54]. Seropositivity is, however, not uncommon and is often interpreted as an indication of previous vaccination since many commercial vaccines incorporate this subtype [12,55].

Cross-protection to EIVs has practical implications in both endemic areas and those free of EI. Due to the logistics associated with the manufacture and licensing of updated commercial vaccines, an ensuing delay may result in novel field strains that have phylogenetically drifted from those EIVs represented in vaccines [3,4,56]. The subsequent mismatch between field and vaccine strains may reduce clinical protection and result in greater virus shedding [52,53,57,58]. Field studies have demonstrated such mismatches, and exposed horses with low antibody titres are far more likely to be responsible for local outbreaks [59]. In addition, poor response to vaccination has also been implicated in driving antigenic drift [60,61,62]. 

Poor responders have been reported in several field studies to EI vaccination, but there have been few experimental studies [55,63,64]. Although several studies have revealed poor responders at a rate of up to 64% (*n* = 11) to ProteqFlu™-Te [55,65,66], this has not been evident in experimental studies or field studies following the 2007 Australian outbreak [42,67]. 

The difficulties of inter and intra HI assay comparisons have been well noted in previous studies, with inter-assay variations of up to 100-fold being reported [11,68]. Inter-batch variation of the HI assay results was confirmed by the laboratory manager at the testing laboratory AAHL (author J.W.). The SRH assay is considered to be more comparable between batches and offers quantitative advantages over the HI [68,69,70]. Single radial haemolysis assays are, however, not available in Australia and may have provided more of a comparable indication of the level of protection that may be correlated with antibody responses. There was a large difference in the median HI levels for the horses in this study and the horses in a concurrent study conducted by the author (C.E., unpublished work). No titre was lower than 64 in the latter group. The number of absent or poor HI antibody responders in this study, 14 days after V1 to both H3N8 antigens, was considered relevant, and if sampling was conducted prior to V2, it would not be affected by the alternative timing of V2 administration. At least three horses remained unavailable for testing at several time points due to logistical vagaries associated with a field study. 

Variables associated with this study that were less controlled than an experimental one included appropriate cold chain logistics and vaccine handling and administration and the reliable identification and presentation of horses to multiple private practitioners. These basic factors could not necessarily be taken for granted, particularly due to spring racing schedules and demanding climatic conditions (outside temperatures exceeding 40 °C at times). Finally, the collection, storage and processing of sera need to be adequate to maintain the integrity of the immunoglobulins being assayed. Reliable freezing of samples is particularly important if prolonged storage precedes processing, as was the case in this study.

It is well accepted that vaccine strains need to be more closely matched to circulating EI viruses [56,71]; however, there are practical considerations involved in regular updates of commercial vaccines. Novel vaccine technologies able to impart heterologous responses reduce the necessity for such close matching and provide a wider margin for protection against emerging EIVs capable of disease outbreaks.

Detection of cellular immune responses in horses following infection or vaccination is complex, in contrast to the measurement of antibodies. Notwithstanding, the significance of CMI in terms of cross-protection and longevity lends impetus to the development of reliable and user-friendly methods of detection. Detection of increased production of IFN-γ serves as a correlate of an upregulated type 1 (cellular) immune response.

Although significant increases in IFN-γ from the PBMCs of the three previously infected (positive control) ponies were detected compared to all other groups, no such difference was apparent in horses vaccinated 6 years previously from the negative control group. These results indicated that the techniques used were specific enough to identify IFN-γ increases in the positive control group; however, analytical sensitivity was not sufficient to identify such a response in horses vaccinated six years previously.

Production of IFN-γ in response to a specific pathogen provides an insight into Type 1 immune responses driven mainly by T helper (Th) cells and cytotoxic lymphocytes (CTLs). These techniques have been developed over the last decade. Detection of specific IFN-γ following EIV infection and, more recently, vaccination has been consistent [24,27,30,37,72].

The technique of prolonged incubation of autologous PBMCs co-cultured with EIV-infected PBMCs is designed to stimulate and enhance the presentation of antigens via MHC class I and II molecules [24]. Such prolonged incubation increases the number of EIV-specific T-cells by stimulating the clonal expansion of antigen-specific or memory T-cells, facilitating detection and subsequent cytokine production [22,44,73]. 

There have been few reports of long-term CMI responses in horses. Detection of a Type 1 cellular immune response to EIV vaccination has only been reported over the last two decades and has been represented by the detection of EIV-specific IFN-γ or its mRNA [18,24,28,74]. Methods to amplify effector T-lymphocytes have been considered necessary to detect cytokine responses [24]. Unlike herpesviruses, EIV infection is transient; hence, there is limited repeat antigenic stimulation [44]. Such limited antigenic exposure to EIV is likely to account for the generally low number of EIV-specific T memory cells in the PBMC population. Although horses in this study were vaccinated some 71 months prior to testing, cell-mediated and humoral responses have been detected following vaccination with recombinant canarypox constructs after a shorter time period [30,31,32,47]. Detection of humoral immunity was unlikely and not expected; however, CMI responses are considered to endure for longer periods [23,75]. Experimental EI challenge studies have demonstrated clinical protection in horses despite low or absent circulating antibodies [13,75]. Protection in these instances is likely to be due to the rapid activation of memory B- and T-lymphocytes. A Type 1 response mediated by IFN-γ is known to drive the CMI component of such immunological responses [5,24,29]. A type 1 immune response mediated by IFN-γ was, however, not detected in the study population. Serial monitoring of vaccinated horses for the detection of IFN-γ as a correlate of (short- and long-term) CMI responses was one of the aims of this study. To attempt this, PBMCs were collected from the eight horses vaccinated with the accelerated regimen; however, a minimum of 10^6^ cells/mL was considered necessary for this assay [24]. This concentration was not consistently evident in the samples, and concerns regarding the potential poor response of thawed cells were considered valid. Given the challenges associated with the detection of type 1 immune responses, these samples were considered to be inadequate for IFN-γ assays. 

Interferon-gamma is produced predominantly by natural killer (NK) cells, CD 4+ and CD8+ lymphocytes. The production from NK cells as part of the innate immune response is non-specific and likely formed the bulk of that produced following stimulation with the medium alone. This non-specific production of IFN-γ is considered separate to that following MHC-restricted antigen presentation [20]. It is possible that a pronounced innate response to the medium alone (serving as a negative control) may have reduced the amplitude of an EIV-specific MHC-restricted response. Relatively greater responses to medium alone were recorded in two vaccinates and one control horse, resulting in a negative value for EIV-specific IFN-γ. Regardless of whether this dampened the EIV-specific response, the statistical power of the study may have been reduced by the amplitude of the non-specific response in 3 of the 15 horses (20%) in the study. 

Methods to measure EIV-specific IFN-γ have included enzyme-linked immune spot (ELISPOT™) and intracellular cytokine staining with flow cytometric cellular enumeration. In addition to identifying IFN-γ, PCR methods to identify the upregulation of gene expression have been successfully reported [28,74]. The ELISPOT technique is well described; however, it is considered, at best, semi-quantitative [44]. Flow cytometric techniques have also been well described for the assessment of EIV-specific IFN-γ production and are considered more quantitative, allowing for cellular enumeration [30,44,72].

Although no significant difference to controls was detected, the upper 95% CI indicated IFN-γ production some 1.3 times greater in the accelerated vaccinates than in controls, which may provide some basis for a larger perspective type study; work is currently underway to identify suitable horses for this. 

Assessment of cellular immune responses in horses remains limited to research laboratories to date. With technological advances, the opportunity for commercial laboratories to assess cellular responses to vaccination would provide more comprehensive and relevant information. Such practical determination of broader immune responses would likely indicate susceptibility or otherwise to a novel circulating EIV strain.

## 5. Conclusions

As vaccine technology is becoming more sophisticated, facilitating more comprehensive immune responses, accurate and reliable assessment of immune responses could enhance the monitoring and containment of EI in horses worldwide. Given the potentially serious consequences of viral shedding in vaccinated horses, an understanding of cross-protective humoral responses and cellular immune responses following vaccination should improve the ability to predict at-risk horses following EI outbreaks.

## Figures and Tables

**Figure 1 vaccines-10-00855-f001:**
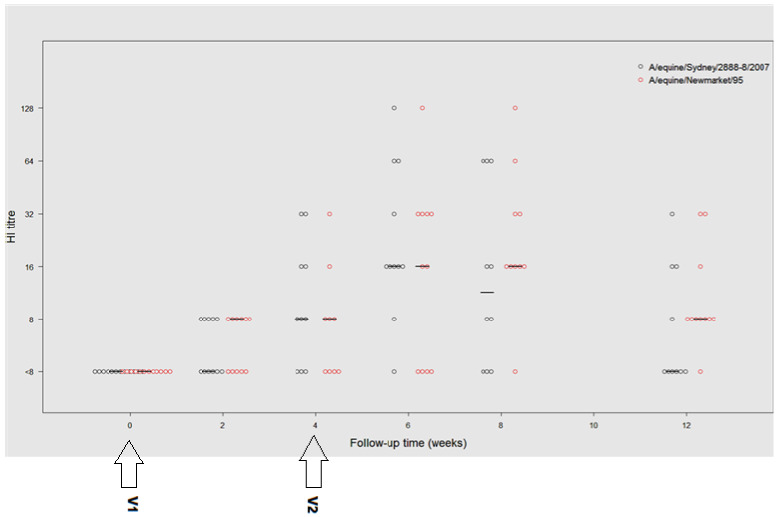
Haemagglutination inhibition (HI) titres to A/equine/Sydney/072007 (blue circles) and A/Equine/Newmarket/95 (red circles) at fortnightly time points following ProteqFlu-Te^®^ vaccination of 12 racehorses (V1 and V2). Minimum recorded dilutions were 1/8 (8). Horizontal lines represent median levels at each time point.

**Figure 2 vaccines-10-00855-f002:**
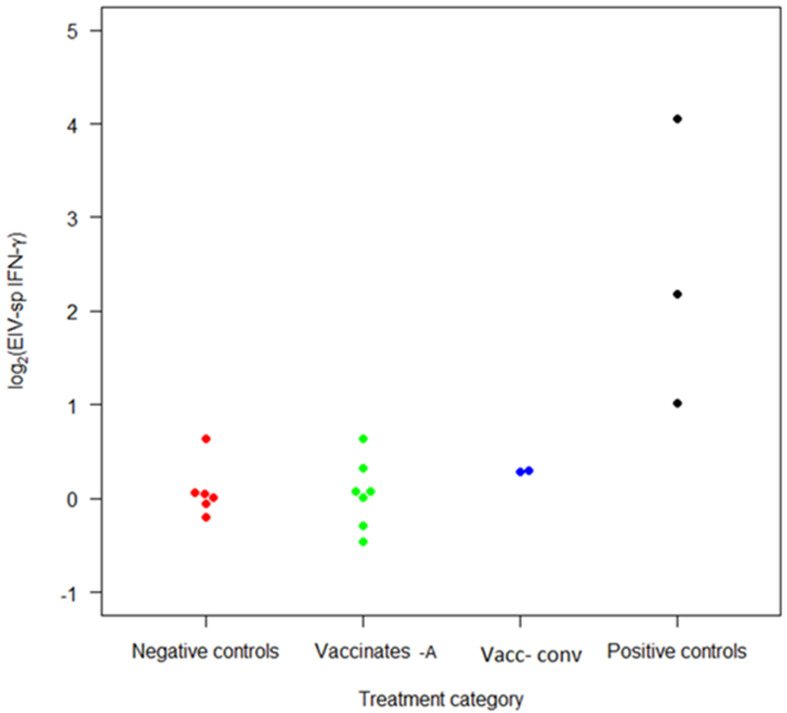
Log 2 interferon-gamma production (IFN-γ) of peripheral blood mononuclear cells from horses and ponies in the study population vaccinated with ProteqFlu-Te™ 6 years prior. EIV-specific IFN-γ following stimulation with A/Equine/Sussex89 (H3N8). Vaccinates-A: vaccinated using accelerated regimen V1-V2 14 days, Vacc-conv: vaccinated using conventional regimen V1-V2 28 days.

**Table 1 vaccines-10-00855-t001:** Equine influenza virus strains used in this project (H3N8 subtypes unless otherwise designated).

EIV	Lineage	Use
A/equine/Newmarket/95	Eurasian	HI assay
A/equine/Newmarket/2/93	Eurasian	HA used in vaccine
A/equine/Sussex/89	Eurasian	γ-Interferon assay
A/equine/Kentucky/94	American	HA used in vaccine
A/equine/Sydney/07	American Fc1	HI assay (local outbreak isolate)
A/equine/Richmond/1/07	American Fc2	γ-Interferon assay
A/equine/Prague/1956 (H7N7) =	N/A	HI assay

**Table 2 vaccines-10-00855-t002:** Study population demographics of racehorses in training and racing from metropolitan and regional racing stables, vaccinated with ProteqFlu™-Te/ProteqFlu™ V1-V2 28-42 days.

Horse ID	Age (Years)	Sex
1	6	Female
2	5	Female
3	4	Gelding
4	5	Stallion
5	8	Gelding
6	3	Female
7	2	Female
8	4	Gelding
9	3	Female
10	3	Female
11	3	Female
12	4	Gelding

**Table 3 vaccines-10-00855-t003:** Haemagglutination inhibition (HI) titres in racehorse sera following vaccination with ProteqFlu-Te^®^ at 0 and 4 weeks.

Horse ID	Day 0 V1 ^a^		Day 14		Day28 V2		Day 42		Day 56		Day 84	
	Sy07b	NM95c	Sy07	NM95	Sy07	NM95	Sy07	NM95	Sy07	NM95	Sy07	NM95
0	<8	<8	8	8	16	16	NA	NA	64	128	16	32
1	<8	<8	<8	<8	8	8	64	128	64	64	8	32
2	<8	<8	8	<8	<8	<8	128	16	64	16	16	8
F2	<8	<8	8	<8	16	<8	32	<8	NA	NA	NA	NA
20	<8	<8	<8	<8	<8	NA	<8	<8	<8	<8	<8	<8
24	<8	<8	<8	NA	NA	8	<8	32	<8	16	32	8
26	<8	<8	NA	8	32	<8	16	32	8	32	8	16
43	<8	<8	<8	8	8	32	8	32	16	32	<8	8
47	<8	<8	<8	8	8	8	64	16	NA	NA	<8	8
126	<8	<8	<8	<8	NA	NA	16	<8	16	16	<8	8
142	<8	<8	8	8	NA	NA	16	32	16	16	<8	8
143	<8	<8	<8	8	<8	NA	16	16	8	16	<8	8

V1 and V2 are the days of the first and second vaccinations, respectively. HI titres against A/equine/Sydney/07 (Sy07) (H3N8). HI titres against A/equine/Newmarket/95 (H3N8) (NM95). ^a^ V1 and V2 are the days of the first and second vaccinations respectively, NA—not available.

**Table 4 vaccines-10-00855-t004:** Group 2 study population of control, vaccinated and EIV-infected horses and ponies.

EIV Status	Number	Breed
ProteqFlu-Te V1-V2 14 day interval	7	Thoroughbred
ProteqFlu-Te V1-V2 42 day interval	2	Thoroughbred
Previously infected	3	Welsh Mountain Pony
Uninfected/unvaccinated	6	Thoroughbred

## Data Availability

All data generated for this study have been provided in the Section 3.

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
