# Peer review of "Assessment of Humoral and Long-Term Cell-Mediated Immune Responses to Recombinant Canarypox-Vectored Equine Influenza Virus Vaccination in Horses Using Conventional and Accelerated Regimens Respectively"

_vaccines, 2022, doi:10.3390/vaccines10060855_

Round 1

Reviewer 1 Report

1.The image in Figure 1 needs to be adjusted.

2.The house ID in Table 2 and Table 3 are inconsistent.

3.References have inconsistent uppercase and lowercase letters.

4.Whether the test animals were free of infection from other diseases?

5.Some typing mistakes were in the manuscript.For example,Page 8 line 249,need to add a space between "14" and "days".It should be corrected throughout the paper.

Author Response

1.The image in Figure 1 needs to be adjusted.

Adjusted  thank you

2.The house ID in Table 2 and Table 3 are inconsistent.

Correct thanks these are different study populations as explained in the materials and methods

3.References have inconsistent uppercase and lowercase letters.

corrected  thank you

4.Whether the test animals were free of infection from other diseases?

All were racehorses in training free from clinical disease we will specify in materials and methods

Some typing mistakes were in the manuscript.For example,Page 8 line 249,need to add a space between "14" and "days".It should be corrected throughout the paper.

corrected  thank you

Reviewer 2 Report

Dr. Charles El-Hage has submitted the manuscript entitled “Assessment of humoral and long-term cell- mediated immune responses to recombinant canarypox-vectored equine influenza virus vaccination in horses using conventional and accelerated regimens respectively” into the journal vaccines. Authors tried to assess both host’s humoral and cell-mediated immune responses to a EI vaccine in racehorses by two different vaccination schedules. But the manuscript failed to answer the mainly questions as shown below:

(i) Any possible relative were present between HI titers and interferon- gamma production in vacationed horses?

(ii) How was the IgA levels in the vacationed horses? Since IgA is a vital local immune protein against viral infection.

(iii) lines 64 to 65, references is wrong?

(iv) no figure 3 (line 239) are shown in this study.

(v) why only two data present in the group vacc-cov?

Author Response

  • Any possible relative were present between HI titers and interferon- gamma production in vacationed horses?

That is a good point and we would have done so had logistics been available. Unfortunately HI was conducted on a separate population (study group 1 ) to the CMI-interferon) study group 2 Also it was anticipated that HI titres would be minimal several years after vaccination when CMI assessment was determined via IFNg levels

  •  

(ii) How was the IgA levels in the vacationed horses? Since IgA is a vital local immune protein against viral infection. Yes absolutely agree that mucosal IgA is integral part of immunity to EI unfortunately it was outside of the scope of the this study looking at systemic responses. Note we have added this point in the introduction segment however

(iii) lines 64 to 65, references is wrong?

No - these references pertain to of heterologous immune responses in horses to EIV my references chosen highlight the important of this cross protection this is covered by Elton and Cullinane  2013(Equine influenza: Antigenic drift and implications for vaccines) and Bryant and Paiilot 2011 Comparison of two modern vaccines and previous influenza infection against challenge with an equine influenza virus from the Australian 2007 outbreak. And Paillot and Prowse 2010  Efficacy of a whole inactivated EI vaccine against a recent EIV outbreak isolate and comparative detection of virus shedding

(iv) no figure 3 (line 239) are shown in this study.

Adjusted to figure 2  thank you

(v) why only two data present in the group vacc-cov?

This group referred to the two horses that received a conventional primary course of 4 weeks. Compared to the main interest group that received an accelerated primary course EIV vaccination regimen, Thanks for your attention to this detail

Reviewer 3 Report

This paper talks about the long-term immune responses after horses took the vaccines for equine influenza. Even though the virus is a horse virus in the veterinary field, I believe this paper has a big significance. However, I would suggest the authors emphasize the significance of their studies more. For example, in introduction, please describe the symptoms of EI and the results of the infection. Is this infection mild, or lethal? Why it is so important to study it? How will the virus impact humans? Financially, or any cases showing transmissions between species? The significance of the study should be emphasized in the introduction. Also, in the discussion part, how could the study be used in the field? Can the result impact a larger scale, like an impact not only restricted to the local area, but also to worldwide?

Below are some small things that need to be changed:

  1. Please make sure on the consistency of the texts. Format of citations: check line 62-70. Format of numbers: check line 178, 271.
  2. Please check line 191. What’s NP? Nucleoprotein? Seems like you first mentioned it in line 162. Please put full name first and then use abbreviations. This will help readers understanding your paper.
  3. Please change your figure 1. Currently it’s a mirror image.
  4. Line 239-240 you said there is a figure 3. However, in your submitted manuscript there’s no figure 3.

Author Response

More than happy to include these suggestions hence introduction has been changed to include these excellent points further discussion about relevance and importance of assessment of comprehensive EIV immune responses was added to the latter part of the discussion-see edited draft

1. Please make sure on the consistency of the texts. Format of citations: check line 62-70. Format of numbers: check line 178, 271.--

citations corrected 

2. Please check line 191. What’s NP? Nucleoprotein? Seems like you first mentioned it in line 162. Please put full name first and then use abbreviations. This will help readers understanding your paper.

good point well accepted and corrected 

3. Please change your figure 1. Currently it’s a mirror image.

oops done apologies 

4. Line 239-240 you said there is a figure 3. However, in your submitted manuscript there’s no figure 3

Apologies for this oversight changed to figure 2